Integrating particle swarm optimization with backtracking search optimization feature extraction with two-dimensional convolutional neural network and attention-based stacked bidirectional long short-term memory classifier for effective single and multi-document summarization

http://orcid.org/0000-0003-2747-3919 Rautaray Jyotirmayee 1 jyotirmayee.1990@gmail.com
Panigrahi Sangram 2
Nayak Ajit Kumar 2
1 Department of Computer Science and Engineering, Siksha O Anusandhan University Institute of Technical Education and Research , Bhubaneswar, Odisha , India
2 Department of Computer Science and Information Technology, Siksha O Anusandhan University Institute of Technical Education and Research , Bhubaneswar, Odisha , India
Alatas Bilal
Electronic publication date: 2024 Dec 12
Publication date: 2024
Volume: 10
Electronic Location ID: e2435
Received 2024 Jul 31; Accepted 2024 Sep 28
Copyright: © 2024 Rautaray et al.
Copyright year: 2024
Copyright holder: Rautaray et al.
License: This is an open access article distributed under the terms of the Creative Commons Attribution License, which permits unrestricted use, distribution, reproduction and adaptation in any medium and for any purpose provided that it is properly attributed. For attribution, the original author(s), title, publication source (PeerJ Computer Science) and either DOI or URL of the article must be cited.
License URL: https://creativecommons.org/licenses/by/4.0/

Keywords: Text summarization, Single document, Multi-document, ABS-BiLSTM, Dimensional convolution neural network, Particle swarm optimization and backtracking search optimization

Funding: The authors received no funding for this work.

==============================
The internet now offers a vast amount of information, which makes finding relevant data quite challenging. Text summarization has become a prominent and effective method towards glean important information from numerous documents. Summarization techniques are categorized into single-document and multi-document. Single-document summarization (SDS) targets on single document, whereas multi-document summarization (MDS) combines information from several sources, posing a greater challenge for researchers to create precise summaries. In the realm of automatic text summarization, advanced methods such as evolutionary algorithms, deep learning, and clustering have demonstrated promising outcomes. This study introduces an improvised Particle Swarm Optimization with Backtracking Search Optimization (PSOBSA) designed for feature extraction. For classification purpose, it recommends two-dimensional convolutional neural network (2D CNN) along with an attention-based stacked bidirectional long short-term memory (ABS-BiLSTM) model to generate new summarized sentences by analyzing entire sentences. The model’s performance is assessed using datasets from DUC 2002, 2003, and 2005 for single-document summarization, and from DUC 2002, 2003, and 2005, Multi-News, and CNN/Daily Mail for multi-document summarization. It is compared against five advanced techniques: particle swarm optimization (PSO), Cat Swarm Optimization (CSO), long short-term memory (LSTM) with convolutional neural networks (LSTM-CNN), support vector regression (SVR), bee swarm algorithm (BSA), ant colony optimization (ACO) and the firefly algorithm (FFA). The evaluation metrics include ROUGE score, BLEU score, cohesion, sensitivity, positive predictive value, readability, and scenarios of best, worst, and average case performance to ensure coherence, non-redundancy, and grammatical correctness. The experimental findings demonstrate that the suggested model works better than the other summarizing techniques examined in this research.

Introduction

A multitude of online materials, such as websites, social media platforms, users feedback, news, and webpages, serve as rich sources of data. Additionally, extensive collections of news stories, publications, legal documents, novels, scientific papers, and biomedical accounts offer a plenteous amount of textual information (El-Kassas et al., 2021). Every day, the dimension of written information on the Internet and other archives increases significantly, making text summarization essential. The rapid advancement of technology has led to the publication of numerous electronic records online, making it challenging for users to identify relevant data and potentially overlook many interesting and important documents. Therefore, a reliable automated summarization method is now crucial. Text summarization is typically categorized into two main types: single document and multi-document summarization. Single-document summarization shortens the content of single document into a compact and concise summary, while multi-document summarization synthesizes information from multiple sources into a cohesive overview (Ma et al., 2022; Ma, 2024). Additionally, summarizing algorithms can be categorized by language use: multilingual algorithms summarize documents in at least two distinct languages, whereas single-language algorithms summarize documents written in only one language. The method of generating a summary can be either extractive or abstractive. Extractive summarization is the process of choosing and rearranging words, phrases, or sentences from the original content. However, abstractive summarizing entails creating new sentences that include the primary ideas and concepts of the original text (Gambhir & Gupta, 2017; Liu et al., 2021).

Single and multi-document summarization can be seen as optimization challenges aimed at creating an ideal summary that encompasses the most important sentences from the unique documents. Nature-inspired optimization methods are particularly effective in tackling this issue. Various metaheuristic techniques have been utilized in both single-document summarization (SDS) and multi-document summarization (MDS), such as particle swarm optimization (PSO), artificial bee colony (ABC), harmony search (HS), cuckoo search (CS), genetic algorithms (GA), ant colony optimization (ACO) (Tomer & Kumar, 2022), and the firefly algorithm *FFA) (Prathap & Rathinasabapathy, 2023). Optimization is vital for raising the performance level and efficiency of text summarization models. Recent research advancements, especially in clustering, evolutionary algorithms, and deep learning, have yielded promising results in this field (Wang, Tan & Liu, 2018; Mirjalili, 2018; Sanchez-Gomez, Vega-Rodríguez & Pérez, 2018). This research presents a unique evolutionary approach, Particle Swarm Optimization with Backtracking Search Optimization (PSOBSA), meant for feature extraction. PSO, motivated by how birds interact with one another, solves problems by continually improving a potential solution based on a predefined quality criterion. The backtracking search algorithm (BSA) enhances PSO by providing mechanisms to backtrack to previous solutions if a more promising area in the search space is found. PSOBSA integrates the exploration capabilities of PSO with the exploitation abilities of BSA, making it a powerful tool for optimization problems (Zaman & Gharehchopogh, 2022). Deep learning techniques are employed for classification in text summarization. Deep learning has revolutionized this field by enabling the development of models capable of understanding and generating human-like summaries (Lombardi & Marinai, 2020). Traditional text summarization techniques relied heavily on manual feature engineering and rule-based methods, which were limited in their ability to capture the nuances of natural language. In contrast, deep learning models, particularly those based on neural networks, can learn to represent text in a way that captures semantic meaning and context. Models such as recurrent neural networks (RNNs), long short-term memory networks, and transformers have been particularly influential. Throughout this context, a novel deep learning technique, ABS-BiLSTM and 2DCNN, is used for summarizing text in each single and multi-document scenarios (Sun & Platoš, 2023).

Following is the structure of the remaining portion of the article: In “Systematic Literature Review”, related recent efforts on document summarizing are examined. “Proposed Framework” provides a detailed description of the suggested approach. A system model that has been mathematically formulated is one of the models that are used in “Result and Discussion” to compare the performance of the proposed model with other models. “Conclusion and Future Work” wraps up the entire article.

Systematic literature review

The following section provides a theoretical study on document summarization using evolutionary algorithms and explores various applications of the deep learning method.

Gidiotis & Tsoumakas (2020) developed a model that divided long documents into sections, processing each section separately. Each section was paired with a specific summarization target, and the resulting partial summaries were combined to create a complete summary. This method required new categorization and machine learning-based section selection for different domains, like financial reports.

Yadav & Sharan (2015) created a model that scored sentences using statistical features (location, Term Frequency-Inverse Document Frequency (TF-IDF), aggregate similarity, centroid) and semantic features, ranking them, and extracting top most ranking sentences to form a concise summary. This dataset was specific to the 2013 Uttarakhand floods.

Debnath, Das & Pakray (2023) used a modified binary Cat Swarm Optimization method, called ETS-MBCSO, for extractive text summarization. The objective function maximized content coverage and informativeness while minimizing redundancy, using sentence length, TF-IDF scores, and similarity measures. However, the system’s accuracy was limited by constraints and the heuristic optimization process.

Saini et al. (2019) established an extractive text summarization system using three multi-objective optimization techniques: self-organizing maps with differential evolution, a grey wolf optimizer, and a water cycle algorithm. These methods clustered sentences semantically and optimized sentence-cluster quality. They noted the need to examine different sentence representation schemes and similarity measures on performance.

Nallapati, Zhai & Zhou (2017) used a RNN to capture sentence sequences in a document. The model employed a single sequence model, reducing the number of parameters. It was trained both extractively, predicting sentence-level probabilities, and abstractively, using a decoder for summary words. The unsupervised approach to converting abstractive summaries to extractive labels was potentially less accurate than supervised methods, which require more annotation.

Shirwandkar & Kulkarni (2018) calculated nine feature values for each sentence to assess relevance, using these to train a restricted Boltzmann machine (RBM) for an initial summary. A second summary was created with fuzzy logic IF-THEN rules. The final summary combined the two using set operations to include the most relevant sentences. This approach was limited by computational time and parameter control.

Xu & Durrett (2019) developed a neural extractive summarization model with syntactic compression by combining extraction and compression techniques. The joint extractive and compressive summarization model (JECS) was trained on labelled data to identify which sentences to extract and how to compress them. The model’s generalizability was affected by the varying compressibility of different datasets.

Rautray & Balabantaray (2017) proposed a new evolutionary framework using CSO to improve MDS. It compared CSO model with PSO and HS based summarizers. The CSO algorithm was computationally intensive and required significant processing power for large datasets.

Zhang et al. (2016) utilized word embeddings for sentence representation and employed an enhanced convolutional neural network (CNN) known as Multiview CNNs (MV-CNNs). MV-CNNs integrated Multiview learning by utilizing multiple convolution filters with different window sizes to analyze sentences from various perspectives. The research faced limitations such as potential increased computational complexity and the need for substantial computational resources to efficiently train the model.

Rezaei, Dami & Daneshjoo (2019) explored deep learning methods for multi-document extractive summarization, comparing deep belief networks (DBN) and Autoencoder neural networks. After normalizing data and extracting features, the feature-sentence matrix was fed into these networks to assess sentence importance based on output scores. They noted limitations such as the potential for overfitting on the training data due to the complexity of the neural networks.

Zhang, Meng & Pratama (2016) adapted the CNN model to perform sentence ranking through regression rather than classification. It utilized a single convolution layer followed by max-pooling on pre-trained word vectors (word2vec), trained end-to-end with these embeddings, avoiding manual feature engineering. The model’s R-2 scores were less satisfactory compared to other systems.

Wang et al. (2020) represented documents as graphs, where nodes denoted words, sentences, and documents, with edges reflecting their relationships. Sentences were linked via shared words to capture semantic connections across the text. The hypergraph neural network (HGNN) comprised graph initializers, a heterogeneous graph layer, and a sentence selector. While the HGNN effectively captured semantic relationships, it might face challenges in highly abstractive summarization tasks.

Table 1 presents research related to single and multi-document text summarization.

Table 1 Research related to single and multi-document text summarization.

Author	Dataset	Preprocessing methods	Advantages	Disadvantages	Results	Metrics	
Gidiotis & Tsoumakas (2020)	ArXiv, PubMed	Sentence segmentation, tokenization, lowercasing, special character removal	Simplicity, noise reduction, compatibility	Information loss, case-sensitive information: may lose distinctions important for proper nouns or acronyms.	The method performed well on short and long document summaries. On longer documents, it outperformed baseline and state-of-the-art approaches. Scalability was good maintaining performance as document length increased.	ROUGE-1, ROUGE-2, ROUGE-L, F-1 score, time complexity	
Yadav & Sharan (2015)	The Hindu, Times of India	Sentence segmentation, stop word removal, stemming, part-of-speech tagging	Noise reduction, improved efficiency, enhanced feature extraction	Potential information loss, language dependency	A hybrid strategy using statistical and sentiment features outperformed standard techniques and provided more cohesive and useful summaries, for opinion-rich literature.	ROUGE-1, ROUGE-2, ROUGE-L, Precision, Recall, F-measure	
Debnath, Das & Pakray (2023)	DUC 2001/2002	Sentence segmentation, tokenization, stop word removal, lowercasing, punctuation removal	Noise reduction, standardization, improved efficiency	Potential loss of meaning and case-sensitive information loss	CSO method summarised single documents competitively and found that their method produced high-quality summaries, beating traditional and state-of-the-art methods.	ROUGE-1, ROUGE-2, ROUGE-L, Precision, Recall, F-measure, Time complexity	
Saini et al. (2019)	DUC 2001/2002	Sentence tokenization, sentence clustering	Multi objective approach, automatic detection, effective summarization	Dependency on clustering, more time complexity	The method outperformed others and its coherent and relevant summaries were assessed by the ROUGE score, precision and recall	ROUGE-1, ROUGE-2, ROUGE-L, Precision, Recall, F-measure	
Nallapati, Zhai & Zhou (2017)	CNN/DailyMail corpora, DUC-2002	Tokenization, word embeddings	Interpretability, no handcrafted features, abstractive training	Data dependency, computational resources	On CNN/Daily Mail and DUC 2002, SummaRuNNer performed well when compared to Lead-3 and feature-rich classifiers.	ROUGE-1, ROUGE-2, ROUGE-L	
Shirwandkar & Kulkarni (2018)	Dataset from Kaggle	Tokenization, sentence embeddings	Accurate sentence extraction and reduction of human efforts for the use of deep learning	The model requires a large and diverse dataset for training to perform effectively and its performance may degrade with domain-specific data	The model outperformed existing models on standard datasets	ROUGE-1, ROUGE-2, ROUGE-L	
Xu & Durrett (2019)	CNN/Daily Mail, New York Times	Tokenization, parsing	The model selects sentences and compresses them by deleting non-essential elements, making summaries shorter while maintaining important information.	Computational complexity and dependency on parsing	The model outperforms baseline extractive systems on datasets and model’s outputs are grammatically correct, outperforming an off-the-shelf compression module.	ROUGE-1, ROUGE-2, ROUGE-L, F-score	
Rautray & Balabantaray (2017)	DUC 2006/2007	Sentence segmentation, tokenization, stop word removal	The use of CSO helps in identifying the relevant sentences while reducing redundancy and maintaining summary cohesiveness outperforming PSO and Harmony search algorithm.	Computational complexity and dataset dependency	The CSO-based summariser outperformed other approaches on DUC datasets and produced non-redundant, readable, and cohesive summaries.	ROUGE scores, F-score, sensitivity, positive predictive value, summary accuracy and inter-sentence similarity	
Zhang et al. (2016)	DUC 2001/2002/2004/2006/2007	Tokenization, lowercase conversion, stop word removal, stemming or lemmatization	A multiview convolutional neural network strategy for multi-document extractive summarisation which may allow the model to summarise distinct perspectives from multiple input sources.	Computational complexity of training multiview CNNs, challenges in generalizing to very diverse document types or domains	The model outperformed extractive summarisation methods in multi-document summarisation on benchmark datasets and it delivered cohesive and meaningful summaries.	ROUGE score, BLEU score, Precision, Recall, F-score	
Rezaei, Dami & Daneshjoo (2019)	DUC 2007	Tokenization, lowercase conversion, stop word removal, punctuation removal, stemming or lemmatization	Here automated feature learning using deep learning models reduces manual intervention and it is effective for handling of large document sets.	Requires large training datasets for optimal performance	Improvements in summarization quality compared to baseline methods	ROUGE-1, ROUGE-2, ROUGE-L, Precision, Recall, F-measure, readability score	
Zhang, Meng & Pratama (2016)	DUC 2001/2002/2004	Tokenization, lowercase conversion, stop word removal, stemming or lemmatization, Sentence segmentation	Ability to capture local patterns and hierarchical features in data, which could be beneficial for identifying important sentences for summary in documents.	Computational complexity of training CNNs, on large document datasets and difficulty in capturing long-range dependencies in text compared to other neural architectures	Well in generating summaries by accurately ranking sentences according to their importance.	ROUGE-1, ROUGE-2, ROUGE-L, Precision, Recall, F-score	
Wang et al. (2020)	CNN/DailyMail, NYT50, and Multi-News	Tokenization, sentence segmentation, stop word removal, stemming or lemmatization	Graph-based summarization, flexibility, inter-sentence relationships	Computational complexity and large datasets are required to train the model effectively	Meant for single-document as well as multi-document summarization and demonstrated strong performance in capturing sentence and document-level relations to improve summary coherence.	ROUGE-1, ROUGE-2, ROUGE-L	

Existing extractive text summarization methods face several limitations, including high computational complexity (Wang et al., 2020), domain specificity (Shirwandkar & Kulkarni, 2018), overfitting (Zhang et al., 2016), reliance on manual feature summarization (Nallapati, Zhai & Zhou, 2017), challenges in balancing multiple objectives and difficulties in processing long documents (Rezaei, Dami & Daneshjoo, 2019). Traditional evaluation metrics like Rouge score may not fully capture summarization quality. Traditional feature extraction technique like TF-IDF has several limitations on text processing. It primarily focuses on the frequency of terms within a document and across a corpus, without considering the contextual relationships between words. This often leads to a lack of semantic understanding, treating words independently, which can result in poor feature representation. Furthermore TF-IDF also generates high-dimensional sparse matrices, leading to inefficiencies in memory usage and computational cost, especially with large corpora. To address these issues novel approaches are addressed.

PSOBSA is used for feature extraction because of its ability to dynamically optimize and select features based on relevance rather than just frequency. As a nature-inspired algorithm, PSOBSA employs a population-based search mechanism that evolves over time, effectively identifying the most relevant features by reducing redundancy and enhancing relevance. This leads to improved model performance. Additionally, PSOBSA’s balance between exploration and exploitation makes it highly efficient in extracting context-aware features, making it particularly suitable for large and complex datasets.

Hybrid deep learning techniques, such as ABS-BiLSTM and 2DCNN, are highly effective for extractive text summarization, particularly for handling larger document sets, which enhances the adaptability of summarization models across various domains and document types. ABS-BiLSTM excels at capturing long-range dependencies and semantic meaning with its bidirectional processing and attention mechanism. In contrast, 2DCNN efficiently extracts local features, such as patterns and important phrases, using convolutional filters. Its parallel processing capabilities ensure faster and more efficient computation, which is useful for large datasets. This combination enhances adaptability, ensuring accurate and computationally efficient summaries by leveraging ABS-BiLSTM’s context understanding and 2DCNN’s feature extraction capabilities.

Objective

The objective of this research is to create a hybrid deep learning framework that uses PSOBSA for feature extraction and 2DCNN and ABS-BiLSTM for classification to provide high-quality summaries for single and multi-document summarization. The model aims to outperform state-of-the-art techniques like PSO, CSO, LSTM-CNN, SVR, BSA, ACO, and FFA in key evaluation metrics like ROUGE-1 (R-1), ROUGE-2 (R-2), ROUGE-L (R-L), precision, recall, F-measure, cohesion, readability, sensitivity, positive predictive value, best case, average case, worst case, and BLEU score. The system also generates coherent, non-redundant, and linguistically accurate summaries to improve information retrieval, decision-making, and efficiency, across disciplines.

Proposed framework

In automatic document summarization, advanced techniques are crucial for creating concise, informative summaries. The preprocessing steps include removing URLs, tokenization, stop-word elimination, stemming, and lemmatization. The method involves feature extraction using an enhanced PSOBSA to gather linguistic, statistical, and semantic data, followed by classification with a 2DCNN and ABS-BiLSTM model. This model focuses on finer-grained fragments and semantic phrases to generate coherent summaries. The approach is evaluated using datasets like DUC 2002, 2003, and 2005 for single-document summarization, and DUC 2002, 2003, 2005, Multi-News and CNN/Daily Mail for multi-document summarization. Figure 1 illustrates the flow of summarization process.

Figure 1 Blueprint for single and multi-document summarization.

Framework of single and multi-document summarization.

Text preprocessing

Text preprocessing converts text into a predictable and analyzable format for specific tasks. This involves removing extraneous information such as URLs, stop-words, and executing tokenization, stemming, and lemmatization, which eliminates irrelevant data and prepare the dataset for further processing (Mridha et al., 2021). The preprocessing steps in text summarization are determined based on the task’s goal, such as extracting relevant features and removing irrelevant information to improve model’s performance. It is crucial for achieving optimal results in text summarization tasks, as they directly impact the quality and accuracy of the generated summaries. In this work, the preprocessing steps are essential for converting raw text into a structured format suitable for feature extraction and classification. These steps, aligned with methodologies in the literature, such as discussed by Gambhir & Gupta (2017) and Nallapati, Zhai & Zhou (2017), help in dimensionality reduction, eliminate noise, and preserve grammatical structure. Stemming removes word suffixes to reduce words to their root forms, often resulting in non-lexical stems, whereas lemmatization considers the context of a word and reduces to its base or dictionary form, improving the quality of features extracted for models. In our approach, the evolutionary algorithm, PSOBSA is employed to optimize the pre-processed text and extract features effectively. Our preprocessing pipeline is specifically tailored to the needs of the ABS-BiLSTM and 2DCNN classifier models used in the summarization system. The framework effectively extracts linguistic, statistical, and semantic features by implementing chosen preprocessing techniques enhancing summarization accuracy and coherence. Figure 2 below illustrates the preprocessing steps.

Figure 2 Sequence of pre-processing steps.

Sequence of pre-processing steps in details.

Removing URL’s

The objectives of data cleaning include eliminating duplicate URLs, retrieving active URLs, preserving URLs that contain HTML elements in the web page’s source code, and including websites with information in English.

Stop word removal

Specifically, common English words like “a,” “an,” “the,” and others are removed from sentences because they have lower centrality within the document. Eliminating these stop words helps to concentrate on the relevant weighted words and significant tokens, as they do not influence in the extraction of key information.

Tokenization

This involves breaking down long texts into manageable chunks, such as dividing sentences into individual words.

Stemming

It entails breaking down words into their “stem,” or root form. The stem is not necessarily a valid word itself, but it captures the core meaning of related words. For example, the stem of words “running,” “runner,” and “ran” could be “run.” Stemming helps in grouping words with similar meanings together, improving the efficiency and effectiveness of text analysis and search algorithms.

Lemmatization

In this stage, input words or tokens are reduced to their lemmas. While stemming simply removes word suffixes, lemmatization reduces words to their base form and ensures correct meaning by using a dictionary. Grammatically, lemmatized words remain meaningful when shortened, which is crucial for a model aiming to provide grammatically correct summaries. For example, instead of the stem “shar” from “sharing,” we obtain the lemma “share”.

Feature extraction using improved PSOBSA

Feature extraction is a pivotal process in machine learning and data analysis, focusing on reducing dimensionality while retaining essential information from raw data. The Enhanced PSOBSA method improves traditional techniques by merging the worldwide search capabilities of PSO with the adaptive refinement mechanisms of BSA. As a result, improved PSOBSA significantly boosts the performance and precision of predictive models across diverse applications (Yadav & Sharan, 2015).

Improved PSO algorithm

The PSO algorithm emulates the social behaviour of birds in a flock and is a population-based search method. In PSO, particles (individuals) move through a hyper-dimensional search space. Their positions are influenced by the social-psychological tendency to mimic others’ successes and their own experiences. Randomly generated particles within the search space symbolize potential solutions to the optimization challenge. Each particle has a position vector and a velocity vector and can remember its previous best position. The swarm collectively searches for optimal solution by updating each particle’s position according to its own and its neighbours’ best positions. Numerous particles are created randomly in the search of the PSO. Every particle in a swarm is a group of particles that together offer a possible solution to the optimization issue (Zaman & Gharehchopogh, 2022). The particle’s speed is changed using each iteration shown in Eq. (1):

(1) Vij(t+1)=Vij(t)+w1φ1(LBestij−Dij(t)+w2φ2)Sbestj−Dij(t).

here i = (1…N) furthermore, j = (1…X), anywhere N and X are the problem dimension and swarm size respectively φ1 and φ2 are two evenly distributed random integers between 0 & 1; w1, w2 are the coefficients of acceleration and Sbest denoted particle i’s best position, Sbest is the swarm’s optimal location globally, and t indicates tth repeating search procedure. Equation (2) updates particle i’s location in each iteration through the addition of the velocity vector to the position vector.

(2) Dij(t+1)=Dij(t)+Fij(t+1).

With inactivity weight application, particle velocity update rule in Eq. (3):

(3) Fij(t+1)=ωFij(t)+w1φ1(LBestij−Dij(t))+w2φ2(SBestj−Dij(t))

ω represents coefficient of inertia weight.

Following the surrounding community’s definition, particle velocity update of Eq. (4):

(4) Fij(t+1)=ωFij(t)+w1φ1(SBestij−Dij(t))+w2φ2(PBestij−Dij(t)).

PBest is the most advantageous location next to particle i. Rings topology, von Neumann topology, and global best version are a few examples of the applicable particle neighbourhood structures in PSO that were developed to improve global search capabilities and prevent local optimum entrapment.

Backtracking search optimization

BSA addresses issues in algorithmic evolution, such as high sensitivity to control settings, long calculation times, and rapid convergence. It has a simple architecture with just two control parameters that manage the rate and amplitude of the search directions matrix, reducing susceptibility to initial control values. BSA uses a stochastic mutation operator that combines crossover, mutation, and selection to generate trial individuals (Deng et al., 2021; Aote et al., 2023). It creates trial communities for robust global exploration and employs two random crossover procedures during the crossover phase. BSA also uses two selection procedures: one for choosing populations from past and present generations and another for selecting the ideal population.

Initialization: The first population is created at random in the consistent search space depicted in Eq. (5) during the initialization step.

(5) Lij∼U(lowj,upj).

Here j = (1, 2, 3,…, X), and i = (1, 2, 3,…, N) where N and X are demographic size and the extent of the issue, in that order. U distribution uniform and each Lij is an individual target in the population (L).

Selection 1: The elderly population is established in Eq. (6):

(6) oldLij∼U(lowj,upj).

Mutation: Eq. (7) uses the mutation process to create the experimental population’s starting form (Deng et al., 2021).

(7) Mutation=L+V×(oldL−L).

Here V regulates the mutation search directions matrix’s amplitude (oldL−L). Since oldL is used compute search-direction, BSA creates the sample for the experiment by artisanal using the experiences of its prior generations.

Crossover: A binary integer-valued matrix (grid) with dimensions in the first step N×X is computed, suggesting that appropriate individuals from the evolutionary population are used to modify the T. The T is then changed If mapij=1, T is efficient by Tij:=Lij.

Selection 2: In selection 2, if Ti It can be more fitness-worthy than the matching Li, it is worn to update Li according to the avaricious choice. The ideal global value of the BSA is going to be revised if the fittest person in the overall population possesses a fitness value that exceeds the ideal global value. Based on these insights, PSOBSA is proposed, an enhanced PSO algorithm by BSA for tackling continual optimization issues.

BSA’s mutation and crossover: To speed up convergence, PSOBSA uses the local best position rather than the historical population from the mutation operator. The mutation operator for the BSA is represented in Eq. (8),

(8) Mutan⁡t=D+V(Pbest−D).

Here V is the matrix of search-direction Pbest amplitude, D is primary position and is an optimum location close to particle i.

(9) Pbest:=permuting(Pbest).

In addition, Pbest the trial position’s ultimate shape, as illustrated in Eq. (9), is produced by the BSA’s crossover procedure, which uses replacement in the trial location.

Procedure of PSOBSA

In PSOBSA, the configuration space consists of key elements such as population size, velocity, position update rules, crossover, and mutation rates. The velocity and position update mechanisms are influenced by both the particle’s personal best and the global best positions, ensuring an effective exploration of the solution space. To leverage PSO’s local search, particle positions are updated using neighbourhood-based PSO, while BSA’s global search is applied through mutation and crossover operators. The steps involved to provide a structured approach for exploring the configuration space while maintaining feasible solutions through effective constraint handling in PSOBSA are: Parameters initialization: The initial population size, velocity, and position of particles are defined. Additionally, inertia weight and acceleration coefficients for velocity updates are specified.

Crossover and mutation rates based on BSA mechanisms are also set.

Representation of scheme: Binary encoding for discrete problems.

Real-valued encoding for continuous optimization problems, where each particle represents a vector of decision variables.

Initial population evaluation: Each particle’s fitness is calculated using the problem-specific fitness function.

Personal best and global best positions are identified in the population.

Updating particle velocity and position: Using Eq. (3), each particle’s velocity is updated based on its personal best position, global best position, and current velocity.

Particle positions are then updated using the new velocity to explore the solution space.

Application of BSA crossover and mutation: Crossover and mutation are applied to introduce diversity within the population.

Crossover combines positions of selected particles, while mutation slightly perturbs the positions.

Fitness evaluation with constraint handling: The fitness of each particle is evaluated.

Penalty function is applied to handle constraints in optimization problems. When a candidate solution (particle) violates a constraint, the penalty function: f′(x)=f(x)+P(x) modifies the fitness value of that solution by adding a penalty, making it less favourable compared to feasible solutions. Where: f(x) is the original fitness function (objective function) that evaluates the quality of the solution x, P(x) is the penalty term added to the fitness function when a constraint is violated and f′(x) is the modified fitness function, incorporating the penalty.

Repair mechanisms are used to adjust any infeasible particles, moving them to a feasible region of the search space.

Backtracking search algorithm (BSA) component: If a particle violates constraints, backtracking is allowed to a previously feasible position, helping to explore a more promising region of the search space.

Adaptive penalty adjustment: Dynamically penalty values are adjusted based on the severity of constraint violations as the algorithm progresses promoting better handling of constraints.

Updating global and personal bests: After fitness evaluation, the personal best for each particle and the global best for the population are updated.

Checking of termination criteria: If the termination criteria are met (e.g., number of iterations or convergence), the algorithm is stopped.

Otherwise, return to step 4, and continue updating particle velocities and positions.

Returning of the optimal solution: The global best position and its corresponding fitness value are output as the optimal solution.

This step-by-step procedure highlights the key operations and mechanisms in the PSOBSA algorithm, particularly focusing on the effective use of the penalty function and backtracking for constraint handling, ensuring a robust approach for solving complex optimization problems which is illustrated in Fig. 3.

Figure 3 Systematic workflow diagram of PSOBSA.

ABS-BiLSTM and 2DCNN

ABS-BiLSTM and 2DCNN offer advanced techniques for single and multi-document summarization. ABS-BiLSTM leverages bidirectional LSTMs with attention mechanisms to capture contextual information from preceding and succeeding tokens in the text. Meanwhile, 2DCNN applies convolutional operations to detect features across two dimensions, enhancing the model’s capacity to comprehend intricate patterns in the data. Together, these methods improve the accuracy and quality of generated summaries (Sun & Platoš, 2023).

ABS-BiLSTM

A stacked bidirectional LSTM neural network that is attention-based forms the basis of the structure. When converting the text into a fixed-length vector participation, the framework first employs the to clean the text’s data, eliminating particular characters and capitalization and keeping just the details that can be recovered as semantics. To understand the syntax of the sentences in this article, we first analyze the characteristics of the text sequence using stacked Bi-LSTM. Subsequently, a self-attention layer is applied to dynamically weight features, highlighting contextual nuances and addressing ambiguous words. Ultimately, a multilayer perspective is used to create the categorization findings.

Embedding layer: With the use of embeddings, words can now be recognized by computers by being converted into vectors. The neural network language model enables neural network and probabilistic language processing for text models by determining probabilistic language model characteristics through neural networks. When a sentence is lengthy, h{C1,C2,…Ci,…Ch}, subsequently use the learned word vector modelling to do a word vector mapping, which creates a word vector matrix from the word sequence {D1,D2,…Di,…Dt}, whereas matrix size is h∗d, in this case, d is the word vector’s dimension.

Stacked BiLSTM neural network layer: A single neural network cannot effectively extract intricate data characteristics. Deep architectures, like the attention-based stacked Bi-LSTM model, leverage the output of the first Bi-LSTM layer as input for the second. This approach more effectively captures intricate properties, thereby enhancing the model’s feature representation capabilities.

First Bi-LSTM layer {D1,D2,…Di,…Dt}, concealed state is mostly transmitted by the first Bi-LSTM layer, where i stands for ith word, the parameters C1, z1 be distributed to the layer’s members. The concatenation of the forward LSTM’s hidden state and the reversed LSTM’s hidden state provides the information for the subsequent Bi-LSTM layer, serving as the output of the first Bi-LSTM layer. Vector of features representation is produced by the second Bi-LSTM layer ti of text by joining hidden states that are forward and backward. ti is a graph that includes implicit deep-level relationships in text, which is essential for increasing the accuracy of categorization. The hidden ti1 state is represented by the following Eqs. (10)–(12):

(10) ti1→=σ(C1[ti−11,di→]+z1)

(11) ti2→=σ(C1[ti−12,di→]+z1)

(12) ti1=ti1→⊕ti1←.

The subsequent formulas determine the stacked Bi-LSTM’s ultimate output in the second layer Eqs. (13)–(15):

(13) ti2→=σ(C2[ti−12,ti1→]+z2)

(14) ti2→=σ(C2[ti−12,ti1→]+z2)

(15) ti2=ti2→⊕ti2←.

Attention layer: After the stacked Bi-LSTM layer, a self-attention mechanism is applied to weigh each word’s context representation vector, creating an attention layer that reflects each word’s significance to sentence semantics. Finally, proportional summation is used to generate the global semantic representation of sentence S.

Step 1: Work out the weight score ei, which calculates how much i-th word contributes to understanding of uncertain terms. Equation (16)’s output hi across fully linked network having an activation purpose of tanh serves as a calculating procedure.

(16) ei=tanh⁡(Cchi+zc)

The calculation of the tanh activation function is Eq. (17):

(17) tanh⁡(d)=ud−u−dud+u−d

where hi is the layer that is hidden the capable of Cc training parameter and the zc term for bias is the state i-th phrase is used to describe a hyperbolic tangential function, which takes any real integer as input, outputs a value among 0 and 1. It’s utilized for processing information and output; closer output is to 1.0.

Step 2: Determine word’s attention weight αi Softmax function is used following normalization eiT and uc dot creation shown in Eq. (18):

(18) αi=exp⁡(eiTec)∑i=0texp(eiTec)

where ec is a learnable context vector with a random start.

Step 3: Equation (19) shows how the calculated weights for attention are applied to hidden layer outputs of stacked Bi-LSTM, or clause vector S.

(19) S=∑i=0tαihi

For phrases having a self-attention mechanism, the representation vector may be computed using the method described above. Below Fig. 4 shows an architecture of ABS-BiLSTM.

Figure 4 Architecture of ABS-BiLSTM.

Output Layer: The focal layer outputs a high-dimensional representation, S, which is used as the feature vector. This feature vector is then fed into the fully connected hidden layer, where the softmax activation function, as described in Eq. (20), converts it into an N-dimensional vector.

(20) a^=Softmax(MLP(S)).

To train the predictive model, compute the cross-entropy loss function (P) between an anticipated label a^ and ground-truth label ai using Eq. (21) as follows:

(21) P(θ)=−∑ai∈a,a^p∈a^a^ilog⁡a^i+(1−ai)log⁡(1−a^i).

θ symbolizes the capability of the training parameter vector. Modifying a parameter that reflects the changing slope of the loss operation, θ.∇θP(θ) minimizes the cross-entropy loss function using a gradient descent optimization technique θ. Equation (22) is the value of the gradient descent algorithm update formula.

(22) θ=γ.∇θP(θ).

γ shows the learning rate.

2D convolutional neural network

The primary task of 2DCNN is to learn probable characteristics and hidden representations from a dynamic feature pool, producing feature maps that are then reduced in geographic dimensions through max pooling. Dropout layers are used to prevent overfitting, and batch normalization layers are incorporated to address potential issues. Additionally, data must be normalized before being fed to the next layer due to the sensitivity of deep learning models to different data types. The mathematical formulation of the convolutional layer in a 2DCNN is represented through Eq. (23).

(23) ai=σi(ci∗di+zi).

where σi shows the function of sigmoid activation and ai symbolizes the result of ith convolution layer. di references to the 2D data that is the input. Similarly, ci indicates the weight of ith the layer of convolution and illustrates the bias element. The output ai, after the combining processes are completed, stores the feature maps. After that, a max pooling approach is used to carry out the pooling procedures. The solution for the max-pooling layers is presented through Eq. (24).

(24) am=maxi,j∈J(ai.j).

where am represents the decreased feature maps obtained from the max pooling layers. To keep the simulation model from overfitting and other problems, the dropout and batch normalization layers are introduced. Additionally, feature map is flattened into a 1-dimensional vector by flatness layer to provide a connection among the subsequent pooling layers impending completely linked layer. Mathematically, Eq. (25) represents the entire linked layer.

(25) af=gi(cif∗am+zif)

here gi indicates function activation. cif then zif indicates bias factors and weight of the fully connected layer, correspondingly. af displays the whole linked layer’s output. Below Fig. 5 shows the architecture of 2DCNN.

Figure 5 Architecture of 2DCNN.

Hybrid module

A hybrid is composite method that combines the results of the 2DCNN and ABS-BiLSTM modules into a single feature vector. Both models are trained hybridly using a joint weight structure. Both models are trained hybridly using a joint weighted matrix. Lastly, every single feature vector is subjected to a sigmoid function described by Eq. (26), which is used to detect TS sequences.

(26) TSdet=σh(C[h2DCNN,hABS−BiLSTM]+z)

where σh stands for the activation function of the sigmoid curve. h2DCNN and hABS−BiLSTM represent the 2DCNN and ABS-BiLSTM models’ final outputs. Likewise, where b is bias factor and indicates joint weight in a hybrid model.

Result and discussion

This section provides details on the outcome measurement, experimental setup, and experimental results. The viability of the proposed initial-phase is assessed within a simulated environment by employing Python software and utilizing datasets. An Intel(R) Core (TM) i5-3470-outfitted computer is utilized for the exam. Furthermore, 12.7 GB of physical memory (RAM) has been added by the OS creator micro software 10 pro.

The outcomes of the experiments are assessed and contrasted with previous models in terms of statistical metrics like BLEU, R, accuracy, precision, cohesion, readability, recall, and F1-score. One metric used to assess classification algorithm’s efficiency is accuracy. It may be described as the proportion of accurate forecasts to all of the predictions made by the model. The equation for accuracy can be written as Eq. (27):

(27) accuracy=TN+TPFN+FP+TP+TN.

Equation (28) is used to determine recall, which is the percentage of applicable examples that have been retrieved out of every one of the pertinent instances (Bewoor & Patil, 2018).

(28) recall=TPFN+TP.

Precision is metric used to evaluate quality of positive made by a classification model, particularly in issues with binary categorization. It calculates the percentage of precise positive forecasts among all the model’s positive predictions (Bewoor & Patil, 2018). The precision, given by Eq. (29), is as follows:

(29) precision=TPFP+TP.

A test’s exactness is evaluated using the F-measure, which may be calculated as the weighted symphonious mean of the test’s correctness and review in Eq. (30).

(30) F1−score=2TP2TP+FP+FN.

Rouge (Recall-Oriented Understudy for Gisting Evaluation) ratings were first presented and have subsequently become widely recognized measures for assessing text summarization systems. The degree of overlap between machine-generated and human-written summaries is used to measure summarization quality (Gambhir & Gupta, 2017). The ROUGE formula is provided by Eq. (31).

(31) ROUGE=∑vj∈(vi)∑S∈Refsummarizes∑n−gram∈SCountmatch(n−gram)∑S∈Refsummarizes∑n−gram∈SCount(n−gram).

The summary’s phrases are coherent because concepts are linked at the sentence and paragraph levels. This promotes a better knowledge of the overall content. The summary ideas choose a subset of Y or a link between sentences in Y, which is displayed as Eq. (32):

(32) Cohesion=fcoh(s)=1−sim(si,sj)i≠j=1,2,…,n.

A subset of s⊂Y is selected by summary readability in a way that maximizes the inter-sentence connections of s chosen from Y. A greater value of fread(s) suggests a greater readability of the summary, which is demarcated as follows: fread(s) measures similarity between si and sj shown in Eq. (33):

(33) Readability=fread(s)=sim(si,sj)i≢j=1,2,…,n.

The produced summary’s word count is measured by BLEU (Bilingual Evaluation Understudy) in comparison to a reference summary. This assessment is precision-based and is derived from the following Eqs. (34), (35):

(34) BLUE=BP∗exp⁡(∑n−1Nwnlog⁡pn)

(35) BP={1ifc>re1−rcifc≤r.

TP signifies the true positive (TP), false positive (FP), true negative (TN), and false negative (FN). BLEU represents Bilingual Evaluation Understudy.

Additionally, for summary evaluation, sensitivity and positive predictive value (PPV) are employed. These equations are used to calculate PPV and sensitivity, as shown in Eqs. (36) and (37).

Sensitivity

(36) Sensitivity=|Truesen||Truesen|+|Referencesum|.

Positive predictive value The proportion of true positives to the sum of true positives and false positives is referred to as the positive predictive value (PPV).

(37) PPV=|Truesen||Truesen|+|Candidatesum|.

Description of dataset

The dataset comprises various collections of documents, including DUC-2002, DUC-2003, DUC-2005, CNN Daily Mail, and the Multi-News dataset. The DUC 2002, 2003, and 2005 databases play a significant role in testing automated text summarizers. These datasets, released by the DUC, were formed by the National Institute of Standards and Technology (NIST) to promote large-scale experimentation among researchers and advance the area of summarization. By providing standardized datasets, DUC facilitates comparative evaluations of summarization algorithms, allowing researchers to assess and compare the effectiveness of different approaches (https://duc.nist.gov).

The multi-news dataset consists of news stories from newser.com and professional human-written summaries of these articles. The dataset includes sources from over 1,500 sites, each appearing at least five times, facilitating the summarization of a variety of documents and summaries (https://www.kaggle.com/datasets/gowrishankarp/newspaper-text-summarization-cnn-dailymail). Table 2 presents single-document dataset description, while Table 3 presents the details of multi-document dataset.

Table 2 Single document dataset (DUC 2002, DUC 2003 and DUC 2005) details description showing set of files, files per set, type of document, source and summary length.

Type of document	Description of dataset	DUC 2002	DUC 2003	DUC-2005	
Single document	Set of files	50	30.0	50	
Files per set	12	20	25	
Type of document	Human-written query and a set of summaries	News articles and their summaries	Human-written query and five human-written reference summaries	
Source	duc.nist.gov	duc.nist.gov	TREC	
Summary extent	112	101	109	

Table 3 Multi document dataset (DUC 2002, DUC 2003, DUC 2005, CNN Dailymail and News) details description showing set of files, files per set, type of document, source and summary length.

Type of document	Description of dataset	DUC-2002	DUC-2003	DUC-2005	CNN/Dailymail	News	
Multi-document	Set of files	50	30	50	312,085	56,216	
Files per set	12	20	25	N/A	N/A	
Type of document	Human-written querying a set of summaries	News articles and their summaries	Human-written query and five human-written reference summaries	News articles with accompanying bullet-point summaries written by human annotators	News articles aggregated from multiple sources with a human-written summary	
Source	duc.nist.gov	duc.nist.gov	TREC	CNN, Daily Mail, and generated by Karl Moritz Hermann and others (available on GitHub)	Various news websites, processed and compiled by Google AI	
Summary length	112	101	109	56 (CNN)/62 (Daily Mail)	263	

Datasets such as DUC, CNN, and Daily Mail for single/multi-document summarization are preferred based on several key factors: Benchmark status: These datasets, particularly DUC (Document Understanding Conference), are widely recognized as standard benchmarks in the text summarization community. Using them allows for direct comparison with a broad range of existing methods, enhancing the comparative value of our results.

Diverse content: CNN and Daily Mail datasets offer a wide variety of news articles covering multiple topics and writing styles. This diversity helps ensure our model’s robustness across different types of content.

Multi-document capabilities: The DUC dataset, especially, provides multi-document summarization tasks for both single and multi-document summarization which are crucial for evaluating our model’s versatility.

Size and quality: These datasets offer a substantial number of document-summary pairs, which is essential for training and evaluating deep learning models.

Temporal range: By using datasets from different years (e.g., various DUC datasets like DUC 2002, DUC 2003 and DUC 2005), we can assess our model’s performance across evolving language patterns and topics.

The selection of hyperparameters for the ABS-BiLSTM and 2DCNN models in Tables 4–6 was based on a systematic optimization process to achieve the best performance. To balance capturing complicated temporal dependencies and computational efficiency, ABS-BiLSTM used 2–4 stacked layers, 50–512 hidden units, and 0.2–0.5 dropout rate to prevent overfitting. For the 2DCNN, 32–256 filters and 3 × 3, 5 × 5, or 7 × 7 filter sizes were chosen to capture local and wider data correlations. Pooling sizes (2 × 2 or 3 × 3) reduced dimensionality without losing key features, and ReLU was employed as the activation function to mitigate vanishing gradient concerns. The combined model used a batch size of 64 to balance training speed and stability and a learning rate of 0.001 with the Adam optimiser for smooth convergence. To optimise model performance across datasets, these hyperparameters were tested and benchmarked against existing methods.

Table 4 ABS-BiLSTM hyperparameters and its corresponding values.

ABS-BiLSTM hyperparameters and its corresponding values where parameters like number of layers, hidden units and dropout rate and their corresponding values are noted down.

Parameters	Values	
Number of layers (Stacked layers)	2–4 layers	
Hidden units (Neurons per layer)	50–512 neurons	
Dropout rate	0.2–0.5	

Table 5 2DCNN hyperparameters and its corresponding values.

2DCNN hyperparameters like number of filters, filter size, pooling size and activation function and its corresponding values.

Parameters	Values	
Number of filters	32–256 filters	
Filter size (Kernel size)	3 × 3, 5 × 5, or 7 × 7	
Pooling size	2 × 2 or 3 × 3	
Activation function	ReLU	

Table 6 Combined model’s (ABS-BiLSTM and 2DCNN) parameters like batch size, learning rate, optimizer, loss_function and their corresponding values.

Parameters	Values	
batch_size	64.0	
learning_rate	00.001	
Optimizer	adam	
loss_function	categorical_crossentropy	

Hyperparameter tuning studies were conducted to identify optimal configurations for learning rate, batch size, number of layers, and filter sizes. Effects on findings were observed: A deeper ABS-BiLSTM model with more than 512 neurons captures more detailed patterns, which may overfit or increase computing costs. Conversely, fewer neurons reduce model generalisation. In deep architectures like ABS-BiLSTM and 2DCNN, dropout rates below 0.2 provide insufficient regularisation, allowing overfitting. In contrast, a dropout rate above 0.5 deactivates too many neurons, preventing the network from learning by discarding crucial features. Underfitting may result. In 2DCNN, smaller kernels capture finer details, while larger kernels focus on broader patterns, hence tests were needed to establish which size improved feature extraction for summarization. Pooling sizes (2 × 2 or 3 × 3) were changed to reduce dimensionality while keeping crucial information. In combined model construction, too high a learning rate prevents model convergence, while too low delays training, making it harder to get optimal outcomes. Batch size (16 or 32) introduces too much variance in gradient estimates, causing unstable learning, while 128 or 256 batches smooth gradients but require more memory and compute. To calibrate the model for the task, parameter changes were carefully examined for positive and negative consequences.

Figure 6 presents a detailed analysis of various performance metrics for the Multi-Document 2002 dataset. Figure 6A plots precision, recall, and F-measure against epochs, highlighting the model’s convergence and effectiveness. Figure 6B offers a performance analysis of the proposed method against existing techniques like PSO, CSO, LSTM-CNN, SVR, BSA, ACO, and FFA focusing on metrics such as recall, precision, and F-measure. Figure 6C demonstrates that the proposed method achieves higher R-1, R-2, and R-L scores, indicating superior summary quality. Figure 6D provides a comprehensive comparison of metrics like cohesion, sensitivity, positive predictive value, readability, and BLEU score across different methods, showcasing their strengths and weaknesses to aid in selecting the most suitable summarization technique.

Figure 6 Multi-document DUC 2002 dataset.

(A) Performance metrics vs. Epoch. (B) Evaluation of the suggested work in comparison to the current approach. (C) R-1, R-2, R-L comparison of proposed work with existing method. (D) Study of performance metrics for proposed work and existing models.

Figure 7 provides a comprehensive analysis of the performance metrics for the Multi-Document 2003 dataset. Figure 7A plots precision, recall, and F-measure against epochs, offering insights into the model’s convergence and summarization effectiveness. Figure 7B compares the proposed method with existing techniques like PSO, CSO, LSTM-CNN, SVR, BSA, ACO, and FFA evaluating their performance based on metrics such as recall, precision, and F-measure. Figure 7C contrasts R-1, R-2, and R-L scores between proposed and existing methods, highlighting the superior summarization quality of the proposed approach. Figure 7D presents a detailed analysis of additional performance metrics, including cohesion, sensitivity, positive predictive value, readability, and BLEU score, providing a thorough understanding of each method’s strengths and weaknesses in multi-document summarization.

Figure 7 Multi-document DUC 2003 dataset.

(A) Performance metrics vs. epoch. (B) Assessment of proposed model in relation to existing techniques. (C) R-1, R-2, R-L comparison of proposed work with existing techniques. (D) Performance evaluation of proposed work vs. existing techniques.

Figure 8 provides a comprehensive analysis of the performance metrics for the Multi-Document 2005 dataset. Figure 8A plots metrics such as precision, recall, and F-score against epochs, showcasing model’s convergence and summarization effectiveness. Figure 8B compares the proposed method with existing techniques, evaluating their performance based on recall, precision, and F-measure. Figure 8C contrasts R-1, R-2, and R-L scores, highlighting proposed method’s superior alignment with human-generated summaries. Figure 8D offers a detailed comparison of additional performance metrics, revealing the strengths and weaknesses of each technique in multi-document summarization.

Figure 8 Multi-document DUC 2005 dataset.

(A) Performance metrics vs. epoch. (B) Assessment of proposed model in relation to existing techniques. (C) R-1, R-2, R-L comparison of proposed work with existing techniques. (D) Performance evaluation of proposed work vs. existing techniques.

Figure 9 provides a detailed analysis of the performance metrics for the multi-document CNN Daily Mail dataset. Figure 9A plots metrics such as precision, recall, and F-score against epochs, showcasing model’s convergence and effectiveness in generating high-quality summaries. Figure 9B compares the proposed method with existing techniques based on recall, precision, and F-measure, revealing their relative effectiveness. Figure 9C contrasts R-1, R-2, and R-L scores, highlighting proposed method’s superior alignment with human-generated summaries. Figure 9D offers a comprehensive comparison of additional metrics like cohesion, sensitivity, positive predictive value, readability, and BLEU score, detailing the strengths and weaknesses of each summarization technique.

Figure 9 Multi-document CNN daily mail dataset.

(A) Performance metrics vs. epoch. (B) Assessment of proposed model in relation to existing techniques. (C) R-1, R-2, R-L comparison of proposed work with existing techniques. (D) Performance evaluation of proposed work vs. existing techniques.

Figure 10 provides a comprehensive analysis of performance metrics for the multi-document news dataset. Figure 10A plots precision, recall, and F-score against epochs, showcasing the model’s convergence and effectiveness in generating high-quality summaries. Figure 10B compares the proposed method with existing techniques (PSO, CSO, LSTM-CNN, SVR, BSA, ACO, and FFA) using metrics such as recall, precision, and F-measure, revealing their relative effectiveness. Figure 10C contrasts R-1, R-2, and R-L scores, highlighting proposed method’s superior performance in aligning with human-generated summaries. Figure 10D offers insights into various evaluation criteria, including cohesion, sensitivity, positive predictive value, readability, and BLEU score, aiding in selecting the most suitable summarization technique based on specific requirements.

Figure 10 Multi-document news dataset.

(A) Performance metrics vs. epoch. (B) Assessment of proposed work with current techniques. (C) R-1, R-2, R-L comparison of proposed work with existing methods. (D) Evaluation of performance metrics for proposed and existing methods.

Figure 11 provides a comprehensive analysis of performance metrics for the single-document 2002 dataset. Figure 11A plots precision, recall, and F-score against epochs, highlighting the model’s convergence and summarization quality. Figure 11B compares the proposed method with existing techniques (PSO, CSO, LSTM-CNN, SVR, BSA, ACO, and FFA) based on recall, precision, and F-measure, revealing their relative effectiveness. Figure 11C contrasts R-1, R-2, and R-L scores, showing proposed method’s superior alignment with human-generated summaries. Figure 11D offers insights into various evaluation criteria, including cohesion, sensitivity, positive predictive value, readability, and BLEU score, aiding in selecting the most suitable summarization technique based on specific requirements.

Figure 11 Single Document DUC 2002 dataset.

(A) Performance metrics vs. epoch. (B) Comparison of proposed work with the existing method. (C) R-1, R-2, R-L comparison of suggested work with current techniques. (D) Investigation of performance metrics for proposed work and existing methods.

Figure 12 provides a comprehensive analysis of performance metrics for the single-document 2003 dataset. Figure 12A plots precision, recall, and F-measure against epochs, illustrating the model’s learning progress and convergence, aiding in fine-tuning and optimizing performance. Figure 12B compares the proposed method with existing techniques (PSO, CSO, LSTM-CNN, SVR, BSA, ACO, and FFA), showing superior performance in accuracy, precision, recall, and F-measure. Figure 12C contrasts R-1, R-2, and R-L scores, demonstrating the proposed method’s effectiveness in producing high-quality summaries. Figure 12D offers a detailed comparison of metrics such as cohesion, sensitivity, positive predictive value, readability, and BLEU score, evaluating the strengths and weaknesses of each summarization technique.

Figure 12 Single Document DUC 2003 dataset.

(A) Performance metrics vs. epoch. (B) Evaluation of the suggested work in comparison to the current approach. (C) R-1, R-2, R-L comparison of proposed work with existing method. (D) Investigation of performance metrics for proposed work and existing methods.

Figure 13 provides a comprehensive analysis of performance metrics for the single-document 2005 dataset. Figure 13A plots precision, recall, and F-measure against epochs, illustrating the model’s learning dynamics and convergence, aiding in optimizing model parameters and fine-tuning performance. Figure 13B compares the proposed method with existing techniques (PSO, CSO, LSTM-CNN, SVR, BSA, ACO, and FFA), highlighting its superior accuracy, precision, recall, and F-measure, demonstrating its effectiveness in classification tasks. Figure 13C contrasts R-1, R-2, and R-L scores, showing proposed method’s consistent outperformance in generating high-quality summaries. Figure 13D offers a detailed comparison of cohesion, sensitivity, positive predictive value, readability, and BLEU score, providing a thorough understanding of each technique’s strengths and weaknesses in document summarization.

Figure 13 Single-document DUC 2005 dataset.

(A) Performance metrics vs. epoch. (B) Comparison of proposed work with the existing method. (C) R-1, R-2, R-L comparison of proposed work with existing method. (D) Analysis of performance metrics for proposed work and existing methods.

In Table 7, the performance of the 2DCNN-ABS-BiLSTM approach is evaluated across multiple iterations on different datasets: DUC-2002, DUC-2003, DUC-2005, Multi-News, and CNN/Daily Mail. The table presents precision, recall, and F-measure scores for each and every dataset at various iterations. Overall, the approach demonstrates consistent improvement across iterations, with precision, recall, and F-measure scores generally increasing with each iteration.

Table 7 Assessment of 2D CNN and ABS-BiLSTM Framework for DUC 2002, DUC 2003, DUC 2005, Multi-News, CNN dailymail in terms of precision, recall and F-score.

Iterations	DUC-2002	DUC-2003	DUC-2005	Multi-News	CNN/Daily mail	
	Precision	Recall	F-measure	Precision	Recall	F-measure	Precision	Recall	F-measure	Precision	Recall	F-measure	Precision	Recall	F-measure	
1	0.60	0.72	0.70	0.68	0.62	0.65	0.70	0.63	0.68	0.78	0.75	0.73	0.78	0.75	0.77	
2	0.64	0.75	0.71	0.72	0.65	0.69	0.73	0.66	0.721	0.801	0.793	0.773	0.79	0.78	0.80	
3	0.65	0.76	0.74	0.75	0.68	0.737	0.78	0.70	0.754	0.839	0.828	0.801	0.80	0.81	0.84	
4	0.70	0.80	0.76	0.79	0.712	0.759	0.80	0.73	0.793	0.876	0.856	0.843	0.81	0.85	0.87	
5	0.78	0.84	0.82	0.83	0.745	0.771	0.82	0.76	0.831	0.914	0.870	0.874	0.84	0.88	0.9	
6	0.80	0.87	0.86	0.86	0.78	0.834	0.85	0.801	0.853	0.943	0.892	0.902	0.88	0.89	0.91	
7	0.86	0.91	0.88	0.901	0.84	0.871	0.88	0.84	0.88	0.975	0.913	0.931	0.91	0.90	0.92	
8	0.89	0.93	0.91	0.934	0.87	0.904	0.90	0.87	0.905	0.98	0.928	0.957	0.93	0.902	0.93	
9	0.94	0.96	0.95	0.951	0.911	0.92	0.92	0.903	0.923	0.989	0.936	0.96	0.961	0.911	0.94	
10	0.99	0.98	0.97	0.97	0.95	0.94	0.94	0.94	0.93	0.993	0.941	0.97	0.981	0.94	0.97	

Table 8 provides a performance evaluation of several techniques for text summarization across dissimilar datasets: DUC 2002, DUC 2003, DUC 2005, Multi-News, and CNN/Daily Mail. Evaluation metrics include R-1, R-2, and R-L scores. Results show that proposed model consistently outperforms other techniques across all datasets and evaluation metrics, achieving the highest R scores overall. This highlights effectiveness of proposed model in generating quality summaries compared to existing methods across diverse datasets and evaluation criteria.

Table 8 Comparative study of the proposed model and existing methods like PSO, CSO, LSTM-CNN, SVR, BSA for DUC 2002, DUC 2003, DUC 2005, Multi-News, CNN dailymail in terms of Rouge-1, Rouge-2 and Rouge-L score.

Techniques	DUC-2002	DUC-2003	DUC-2005	Multi-News	CNN/Daily Mail	
	R-1	R-2	R-L	R-1	R-2	R-L	R-1	R-2	R-L	R-1	R-2	R-L	R-1	R-2	R-L	
PSO	0.46	0.423	0.56	0.52	0.54	0.32	0.34	0.51	0.55	0.54	0.51	0.36	0.46	0.53	0.49	
CSO	0.421	0.53	0.35	0.31	0.56	0.49	0.456	0.362	0.54	0.33	0.54	0.43	0.57	0.51	0.47	
LSTM-CNN	0.5	0.324	0.47	0.435	0.35	0.45	0.51	0.47	0.531	0.43	0.57	0.53	0.543	0.33	0.52	
SVR	0.302	0.46	0.351	0.539	0.55	0.53	0.49	0.46	0.43	0.49	0.53	0.47	0.493	0.43	0.46	
BSA	0.435	0.51	0.49	0.51	0.43	0.55	0.47	0.53	0.33	0.51	0.43	0.35	0.39	0.49	0.491	
ACO	0.5	0.51	0.53	0.52	0.49	0.5	0.52	0.51	0.53	0.49	0.48	0.53	0.45	0.53	0.51	
FFA	0.49	0.5	0.47	0.49	0.52	0.49	0.46	0.5	0.51	0.46	0.45	0.5	0.5	0.45	0.5	
Proposed	0.59	0.58	0.589	0.6	0.61	0.581	0.584	0.596	0.613	0.588	0.591	0.587	0.6	0.595	0.581	

The proposed model, PSOBSA with 2DCNN and ABS-BiLSTM, offers several advantages over existing techniques for both single and multi-document summarization. By employing PSOBSA, it enhances feature extraction efficiency, improving performance by combining the exploration of PSO with the exploitation of BSA. This hybrid optimization is crucial for accurate feature extraction in summarization tasks. The model’s classification ability is significantly improved by using a 2DCNN and ABS-BiLSTM, allowing it to capture complex text patterns and context more effectively than traditional methods. In single-document summarization, it surpasses methods like PSO, CSO, LSTM-CNN, SVR, BSA, ACO, and FFA across various metrics such as R-1, R-2, and R-L, consistently achieving higher precision, recall, and F-measure scores, sensitivity, readability, positive predictive value, best case, average case, worst case and BLEU score. For multi-document summarization, it outperforms methods like PSO, CSO, LSTM-CNN, SVR, BSA, ACO, and FFA, delivering superior results on datasets like DUC 2002, DUC 2003, DUC 2005, Multi-News, and CNN/Daily Mail with better cohesion, sensitivity, readability, positive predictive value, best case, average case, worst case, Rouge scores and BLEU score. The model’s versatility across multiple benchmark datasets demonstrates its robustness, consistently delivering improved summaries in both single and multi-document tasks. Our proposed model shows performance across the given various metrics, making it a comprehensive solution for generating coherent, accurate, and grammatically sound summaries.

Conclusion and future work

In conclusion, the proposed model introduces a hybrid deep learning framework that integrates PSOBSA for feature extraction and 2DCNN combined with ABS-BiLSTM for classification. The model undergoes comprehensive evaluation on DUC 2002, 2003, 2005 dataset for single document and DUC 2002, 2003, 2005, CNN/Daily Mail, and Multi-News for multi-document summarization tasks. The results demonstrate that the proposed approach surpasses existing state-of-the-art techniques such as PSO, CSO, LSTM-CNN, SVR, BSA, ACO and FFA. Notably, the model achieves higher scores in key metrics, including ROUGE-1, ROUGE-2, ROUGE-L, precision, recall, F-measure, cohesion, sensitivity, readability, positive predictive value, best case, average case, worst case and BLEU score, across the given datasets. The findings underscore the effectiveness of using PSOBSA for feature extraction and the combined 2D CNN and ABS-BiLSTM architecture in generating high-quality summaries that are coherent, non-redundant, scalable and grammatically accurate. High-quality summaries are crucial for retrieving efficient information, saving time, improving efficiency and information accessibility that results in valuable decision-making processes, across various domains. The superior performance of our proposed model suggests its potential to advance in the field of automatic text summarization.

The study has several limitations that future research could address. First, the model is primarily evaluated on English-language datasets, limiting its applicability in multilingual environments, and thus needs to be extended for multilingual summarization tasks to increase its utility. Additionally, the focus on datasets such as DUC, CNN/Daily Mail, and Multi-News may not fully represent the diversity of real-world texts, which restricts the model’s generalizability across various domains. Expanding the evaluation to include more diverse and domain-specific datasets would improve its robustness. Future work can explore hybrid approaches that combine both extractive and abstractive summarization methods to further enhance the quality of the generated summaries. Additionally, incorporating reinforcement learning techniques can improve the coherence and overall quality of the summaries, pushing the boundaries of what can be achieved in automated document summarization.

Supplemental Information

Supplemental Information 1 DUC 2002, DUC 2003 and DUC 2005 datasets.

Supplemental Information 2 Python code for single and multi document.

Additional Information and Declarations

Competing Interests

Author Contributions

Data Availability

The authors declare that they have no competing interests.

Jyotirmayee Rautaray conceived and designed the experiments, performed the experiments, analyzed the data, performed the computation work, prepared figures and/or tables, and approved the final draft.

Sangram Panigrahi analyzed the data, authored or reviewed drafts of the article, and approved the final draft.

Ajit Kumar Nayak analyzed the data, authored or reviewed drafts of the article, approved the final work, and approved the final draft.

The following information was supplied regarding data availability:

The data is available at Kaggle:

- https://www.kaggle.com/datasets/rmisra/news-category-dataset

- https://www.kaggle.com/datasets/gowrishankarp/newspaper-text-summarization-cnn-dailymail.

The DUC datasets are available at: https://duc.nist.gov.

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
