# Peer review of "Integrating particle swarm optimization with backtracking search optimization feature extraction with two-dimensional convolutional neural network and attention-based stacked bidirectional long short-term memory classifier for effective single and multi-document summarization"

_PeerJ Computer Science, doi:10.7717/peerj-cs.2435_

## Round 0.1 · original submission · Major Revisions

Dear authors,

The reviews for your manuscript are included at the bottom of this letter. We ask that you make necessary changes and additions to your manuscript based on those concerns and criticisms. Furthermore, adding a discussion for synthesis of findings, implications, future research, and limitations will be better. Furthermore, the reason for selecting the particle swarm optimization algorithm and backtracking search optimization algorithm for hybridization is not discussed. Configuration space of evolutionary algorithms should be detailed. It should be more specific and comprehensive. Representation scheme (encoding type) and fitness function with constraint functions should be clearly provided. How constraints (for example: for decision variables) are handled should also be provided.

Best wishes,

Reviewer 1 ·

Basic reporting

All comments have been added in detail to the last section.

Experimental design

All comments have been added in detail to the last section.

Validity of the findings

All comments have been added in detail to the last section.

Additional comments

Review Report for PeerJ Computer Science
(Integrating PSO-BSA feature extraction with 2DCNN and ABS-BiLSTM classifier for effective single and multi-document summarization)

1. Within the scope of the study, a new optimization is proposed in the feature extraction phase and a deep learning-based model is proposed in the classification phase in order to perform single and multi document summarization processes using open source datasets belonging to different years.

2. In the introduction, various online materials, the importance of summarization processes, some optimization and classification techniques used in the literature for text summarization, the importance of the subject and the functioning of the article are mentioned. This section is sufficient at a basic level.

3. The literature section mentioned in the second section definitely needs to be organized and detailed. In this section, it is suggested to add a table consisting of sections such as "used dataset, text preprocessing methods, advantages, disadvantages, results, metrics" depending on the problem addressed in the literature (single/multi document summarization). In addition, immediately after this, in order to emphasize the importance of the subject more clearly, the difference of this study from the literature, its basic contributions to the literature and its originality points should be detailed in items.

4. Although the dataset used in the study is suitable for the problem addressed (text summarization), it should be explained more clearly why datasets such as DUC / CNN / Daily Mail are preferred for single/multi document summarization, despite the wide variety of datasets in the literature.

5. When the summarization process and text preprocessing steps in Figure-1 are examined, it is observed that they are sufficient. However, since the selection of preprocessing techniques/steps in this section significantly affects the results, this section should be compared in more detail with the literature and how the preprocessing steps are determined should be discussed more clearly.

6. When the Particle Swarm Optimization with Backtracking Search Optimization technique, procedure and workflow improved for feature extraction are examined in detail, it is observed that it has the potential to contribute to the literature and constitutes an important part of the study.

7. For the classification phase, deep learning-based CNN and LSTM models and hybrid structure were preferred in the study. When the model architectures are examined, it is observed that the architectures are very simple, but their positive effect on problem solving is clearly understood. Here, although the models contain a certain level of originality, clearer explanations and justifications should be made regarding the selection of hyperparameters in tables-3,4 and 5. For example, how were the optimizer, leraning rate, batch size values determined, were different experiments performed? Since the changes in the parameters here may have a positive/negative effect on the results, analyze this section in more depth and provide the justification.

8. The evaluation metrics obtained for the analysis of the results are at a suitable level, albeit at a borderline level, in terms of both type, diversity and comparison. However, here, in order for the proposed model to stand out more, it is definitely recommended to compare the results with at least a few different state-of-the-art methods.

As a result, although the method proposed in the study for text summarization is interesting, all the sections mentioned above should be addressed one by one and answered, and it is definitely recommended that updates be made in the relevant places in the paper.

Reviewer 2 ·

Basic reporting

- The figures need to have proper labeling and clear lines to ensure they are easily understood when viewed in black and white.

The objective of this research is to develop an automatic extractive summarization system for both single and multi-document summaries. However, the conclusion and discussion sections do not address the achievements related to both aspects.

- This article presents many results, but it fails to summarize or conclude the findings in a way that provides a useful conclusion.

Experimental design

This paper includes many comparisons experiments but fails to produce a conclusion that aligns with the objectives.

Validity of the findings

The experiment results has to summarize properly in order to make it a useful information, for example which model is good for single or multi-documents?

---

## Round 0.2 · accepted · Accept

Dear authors,

One of the original reviewers did not respond to the invitation for reviewing your revised manuscript. The other reviewer thinks your paper can be accepted. I also think that the paper has been sufficiently improved. As such, the article is considered acceptable.

Best wishes,

Reviewer 1 ·

Basic reporting

All comments have been added in detail to the last section.

Experimental design

All comments have been added in detail to the last section.

Validity of the findings

All comments have been added in detail to the last section.

Additional comments

Review Report for PeerJ Computer Science
(Integrating PSO-BSA Feature Extraction with 2DCNN and ABS-BiLSTM Classifier for Effective Single and Multi-Document Summarization)

When the responses given to the reviewer comments are examined in detail, they are limited in some aspects, but they are generally at an appropriate level. For this reason, I recommend that the paper be accepted.